# The Magnetic Properties and Magnetocaloric Effect of Pr_0.7_Sr_0.3_MnO_3_ Thin Film Grown on SrTiO_3_ Substrate

**DOI:** 10.3390/ma16010075

**Published:** 2022-12-21

**Authors:** Bojun Zhao, Xiaojie Hu, Fuxiao Dong, Yan Wang, Haiou Wang, Weishi Tan, Dexuan Huo

**Affiliations:** 1Key Laboratory of Novel Materials for Sensor of Zhejiang Province, Institute of Material Physics, Hangzhou Dianzi University, Hangzhou 310018, China; 2All-Solid-State Energy Storage Materials and Devices Key Laboratory of Hunan Province, College of Information and Electronic Engineering, Hunan City University, Yiyang 413002, China; 3Key Laboratory of Soft Chemistry and Functional Materials, Ministry of Education, Department of Applied Physics, Nanjing University of Science and Technology, Nanjing 210094, China

**Keywords:** magnetic properties, magnetocaloric, manganite, film, Pr_0.7_Sr_0.3_MnO_3_

## Abstract

The magnetic behaviors and magnetocaloric effect (MCE) of Pr_0.7_Sr_0.3_MnO_3_ (PSMO-7) film grown on a (001) SrTiO_3_ single-crystal substrate by a pulsed laser deposition (PLD) were studied in this paper. X-ray diffraction with a high resolution (HRXRD) measurement shows that PSMO-7 film is grown with a (001) single orientation. The magnetic properties and the MCE related to the ferromagnetic (FM) phase transition of the PSMO-7 film are investigated using the temperature dependence of magnetization M(T) and the magnetic field dependence of magnetization M(H). The M(T) data suggest that with decreasing temperatures, the PSMO-7 film goes through the transition from the paramagnetic (PM) state to the FM state at around the Curie temperature (T_C_). The T_C_ (about 193 K) can be obtained by the linear fit of the Curie law. Magnetic hysteresis loop measurements show that the PSMO-7 film exhibits the FM feature at temperatures of 10, 100, and 150 K (low magnetic hysteresis can be found), while the film reveals the PM feature with the temperature increased up to 200 and/or 300 K. The research results of M(H) data are consistent with the M(T) data. Furthermore, the magnetic entropy change ($-\Delta S_{M}$) of the PSMO-7 film was studied. It was found that the maximum value of ($-\Delta S_{M}$) near T_C_ reaches about 4.7 J/kg·K under the applied field change of 20 kOe, which is comparable to that of metal Gd ($-\Delta S_{M}$ of 2.8 J/kg K under 10 kOe), indicating the potential applications of PSMO-7 film in the field of magnetic refrigeration.

## 1. Introduction

Perovskite compounds R_1−x_A_x_MnO_3_ (in which R are trivalent ions, such as Pr^3+^, Nd^3+^, and La^3+^, and A are divalent ions, such as Ca^2+^, Sr^2+^, and Ba^2+^) are interesting in the fields of material physics and applicability because these compounds show the interesting electrical transport and magnetic properties [1,2,3,4,5,6,7,8]. The R-site doping by divalent ions, including Ca^2+^, Sr^2+^, and Ba^2+^ usually leads to mixed valence (Mn^3+^ and Mn^4+^) manganite ions, resulting in the emergence of ferromagnetic (FM) double-exchange interactions. Many related studies have indicated that the magnetocaloric effect(s) (MCE) in perovskite compounds take place around the FM to paramagnetic (PM) phase transition [9,10,11,12,13] and the magnetic properties of manganite depend on the R/A-site ionic radii and the ratio of Mn^3+^/Mn^4+^ ions [14,15,16,17,18]. In addition, phase transition plays an important role in these physical properties of perovskite manganite. The first-order phase transition is combined with a structural change. The relatively large magnetic hysteresis and thermal hysteresis can be found in the materials with the first-order phase transition, whereas they cannot be observed in the materials with the second-order phase transition. The materials with the second-order phase transition usually exhibit a relatively small magnetic hysteresis and have potential applicability in the field of magnetic refrigeration.

In perovskite manganite R_1−x_A_x_MnO_3_, the manganite with the doping concentration x = 1/3 usually exhibits the FM to PM transition and the obvious MCE [13,14,15,19,20]. The electrical transport and large magnetoresistance effects of Pr_0.7_Sr_0.3_MnO_3_ (PSMO-7) manganite have been investigated [21,22]. The PSMO-7 exhibits the FM-PM transition at Curie temperature T_C_ = 255 K. The negative magnetic entropy change ($-\Delta S_{M}$) of about 6.3 J/kg K is obtained under a 50 kOe field near T_C_ (235 K) in the nanoscale Pr_0.7_Sr_0.3_MnO_3_ compound [20]. Meanwhile, under the 50 kOe field, the Pr_0.63_Sr_0.37_MnO_3_ single crystal also exhibits a ($-\Delta S_{M}$) value of about 8.5 J/kg K near 305 K [23]. Perovskite manganite PSMO-7 may be one candidate considered for applicability in the field of magnetic refrigeration. However, detailed studies on the MCE of PSMO-7 film have not been reported. Therefore, the PSMO-7 film on the SrTiO_3_ substrate was grown to investigate its magnetic properties and MCE in this work. Remarkably, the FM to PM transition and MCE can be found from M(T) curves, M(H) loops, and M(H,T) data for the PSMO-7 film. Compared to other similar research studies, the PSMO-7 film on the SrTiO_3_ substrate shows the second-order FM to PM transition. Under the 20 kOe applied field change, the maximum value of entropy change reaches 4.7 J/kg K, which is comparable to that of metal Gd. The low magnetic hysteresis and high entropy changes of the PSMO-7 film highlight its potential application in the field of magnetic refrigeration.

## 2. Experimental Details

A 20-nm-thick PSMO-7 thin film was deposited on a (001) single crystal SrTiO_3_ substrate using the pulsed laser deposition (PLD). The detailed film growth process and specific parameters can be found in a previous report [21]. The thickness of the PSMO-7 thin film was dominated by the deposition time. X-ray diffraction with high resolution (HRXRD) was used to study the structure of the PSMO-7 thin film at the Beijing Synchrotron Radiation Facility. The wavelength of HRXRD was 0.1536 nm, and the energy resolution of HRXRD was 4.4 × 10^−4^. The magnetic properties and MCE of PSMO-7 film were investigated by the magnetization versus temperature M(T) curves, magnetic hysteresis M(H) loops, and the isothermal magnetization M(H,T) data, which were performed at the physical property measurement system (PPMS).

## 3. Results and Discussion

Figure 1 shows the HRXRD pattern of the PSMO-7 film on (001) SrTiO_3_. This scan was in the *ω*/2θ mode in the reciprocal space of the SrTiO_3_ (002) plane.

In the *ω*/2θ scan, i.e., during scanning, the speed of the detector was twice that of the sample. This scanning just corresponds to the direction of scanning of an inverted lattice vector in the reciprocal space, so it is usually called radial scanning.

In Figure 1, only reflection peaks for the (002) plane of SrTiO_3_ and the PSMO-7 thin film can be observed, indicating that the PSMO-7 film is grown with (001) orientation. This method for determining the orientation of thin films has been used in many literature studies [17,21]. The full width at half maximum (FWHM) of the diffraction peak for the (002) plane of the PSMO-7 film reveals a value of 0.25°. The above results prove that PSMO-7 grown on SrTiO3 has relatively well growth quality. The crystallite size (d) of the sample can be calculated by using the Scherrer formula, d = 0.89λ/βcosθ, where λ is the X-ray wavelength (0.1536 nm), β is the FWHM of the peak, and θ is the Bragg’s angle. The calculation results show that the crystallite size is about 118 nm. In addition, the PSMO-7 film shows a smooth surface with the rms roughness of about 0.35 nm, which is obtained by an atomic force microscope (not shown here). The detailed magnetism and MCE of the PSMO-7 thin film are further studied in subsequent sections.

Figure 2 shows the magnetization versus temperature M(T) curves of the PSMO-7 film under zero-field-cooling (ZFC) and field-cooling (FC) conditions from 10–350 K at a magnetic field of 100 Oe.

There is a FM to PM transition at the Curie temperature (denoted as T_C_) from the FM metal state at a low temperature to the PM insulator state at a high temperature. The magnetization of PSMO-7 thin film decreases with increasing temperature, as shown in Figure 2, which is a feature of the FM to PM transition. The separation of ZFC and FC M(T) curves at low temperatures is ascribed to the spin glass state and/or magnetic phase inhomogeneity in the film [16,17]. Figure 3 shows the magnetization versus temperature M(T) and the inverse susceptibility versus temperature 1/χ(T) curves measured from the PSMO-7 film in the temperature range of 10–350 K under the FC condition at the magnetic field of 100 Oe.

It can be found that the 1/χ(T) curve can obey the linear relationship described by the Curie law in the high-temperature region of T > T_C_, and the value of T_C_ is determined to be about 193 K. This also demonstrates the PSMO-7 film undergoes a FM to PM transition with increasing temperature.

In order to discuss the magnetic properties of PSMO-7 film, the magnetic hysteresis M(H) loops at temperatures of 10, 100, 150, 200, 250, and 300 K are measured as shown in Figure 4.

Figure 4a reveals the magnetic data without subtracting the diamagnetic background of the SrTiO_3_ substrate, while Figure 4b shows the magnetic data obtained after deducting the substrate’s diamagnetic background. The PSMO-7 film presents the FM to PM transition behaviors, which are consistent with the M(T) data in Figure 3. As seen in Figure 4b and the inset of Figure 4b, at temperatures of 10–150 K, the magnetization rises quickly with the small magnetic field and the M(H) loops can also be observed, suggesting the existence of the FM state. At temperatures above 200 K (e.g., 200–300 K), the M(H) loops gradually disappear and the loops seem to be linear, demonstrating the appearance of the PM state. The M(H) curves significantly oscillate at a temperature of 10 K compared to those at a higher temperature. The oscillations of these data are related to the testing principle of PPMS. Within the high field, the measured film vibrates violently. In this case, at a low temperature (10 K), the violent vibration can cause data oscillation. Figure 5 shows the temperature dependence of saturation magnetization (M_S_) and coercive field (H_C_). With the increase in temperature from 10 K, the M_S_ and H_C_ gradually decrease.

Above 200 K, the values of M_S_ and H_C_ are zero. This phenomenon is consistent with the FM phase transition of the PSMO-7 film, as shown in Figure 2, Figure 3 and Figure 4. It should be noted that, here, the drop in magnetization with the temperature (Figure 5) is related to the transition from the FM to the PM state.

The magnetization versus magnetic field measurements (in the region of 0–20,000 Oe fields) were performed for different temperatures of 10–210 K (with an interval of 5 K) to study the magnetic entropy. Figure 6 exhibits the isothermal magnetization M(H, T) curves of the PSMO-7 film.

The M(H, T) curves were corrected by deducting the diamagnetic background induced by the SrTiO_3_ substrate. Figure 7 shows the Arrott plots obtained from M(H, T) as the M^2^ vs. H/M are plotted to investigate the phase transition types. The shapes of M(H, T) curves at around T_C_ indicate a typical second-order transition, as found in La_1-x_Ba_x_MnO_3_ manganite [16].

The positive slope of the Arrott curve at around T_C_ represents the second-order phase transition, while the negative slope of the Arrott curve at around T_C_ represents the first-order phase transition. It can be seen that the slope is positive near T_C_ for PSMO-7 film, confirming that the film undergoes a second-order phase transition based on Banerjee’s criteria [24].

The temperature dependence of the magnetic entropy change was carried out for studying the MCE of PSMO-7 film. In view of Maxwell’s relation, the magnetic entropy change ($\Delta S_{M}$) value can be obtained via calculating the M (H, T) curves, such as Equation (1),
(1)$$\Delta S_{M}=\int_{0}^{H}\left(\frac{\partial M}{\partial T}\right)dH$$

Equation (1) can be numerically approximated as Equation (2),
(2)$$\Delta S_{M}=\sum_{i}\frac{M_{i+1}\left(T_{i+1},H_{i}\right)-M_{i}\left(T_{i},\;H_{i}\right)}{T_{i+1}-T_{i}}\Delta H$$

In which $M_{i}\left(T_{i},\;H_{i}\right)$ and $M_{i+1}\left(T_{i+1},H_{i}\right)$ are the moment values at $T_{i}$ and $T_{i+1}$ with the applied magnetic field $H_{i}$, respectively [16]. The temperature dependencies of negative entropy changes ($-\Delta S_{M}$) at different magnetic fields of 5, 10, 15, and 20 kOe are exhibited in Figure 8.

As expected, there is a maximum value of ($-\Delta S_{M}$) to be observed in the temperature range of the FM to PM transition (near T_C_) and it increases with respect to the incremental magnetic field. In Figure 8, the position of the maximum value of ($-\Delta S_{M}$) is almost unchanged, which is another confirmation of the second-order phase transition. An obvious increase in the ($-\Delta S_{M}$) value appears at around T_C_. Under a relatively high magnetic field (20 kOe), the film vibrates violently, resulting in a fluctuation of magnetic data. Due to the ferromagnetic transition, the value of 4.7 J/kg K can be found at the Curie temperature (it can be found that the data are fluctuating). Near the Curie temperature, the ($-\Delta S_{M}$), called the conventional MCE of the PSMO-7 film, is attributed to the spin alignment in a specific field direction. The ($-\Delta S_{M}$) value of 4.7 J/kg K at around 190 K can be obtained for the applied field change of 20 kOe. This value of the entropy change for PSMO-7 film grown on the SrTiO_3_ substrate is comparable to that of the metal Gd ($-\Delta S_{M}$) of 2.8 J/kg K under a 10 kOe field [25]). Compared with other perovskite manganites, the PSMO-7 film exhibits a significantly low field magnetocaloric effect [13,14,15,19,20,21,22,23]. This may be related to the relatively high growth quality of the film. The high growth quality of the film is conducive to the enhancement of the double exchange interaction, which leads to the improvement of the significant magnetocaloric effect. Moreover, the low magnetic hysteresis can be found in the PSMO-7 film. These research results demonstrate the potential applications of PSMO-7 film in the field of magnetic refrigeration.

## 4. Conclusions

The PSMO-7 film has been successfully grown on the (001) SrTiO_3_ substrate. The magnetic properties and MCE of the film were studied. The Pr_0.7_Sr_0.3_MnO_3_ film shows the second-order FM to PM transition at around T_C_ (193 K). Under 20 kOe of applied field change, the maximum value of ($-\Delta S_{M}$) reaches 4.7 J/kg K, which is comparable to that of metal Gd. The low magnetic hysteresis and high entropy change of PSMO-7 film highlight its potential applications in the field of magnetic refrigeration. The present work on the magnetic properties and MCE of PSMO-7 film on SrTiO_3_ is of great importance because it can deepen the understanding of magnetic properties; the high entropy change and low magnetic hysteresis are useful for practical applications in magnetic refrigeration.

## Figures and Tables

**Figure 1 materials-16-00075-f001:**
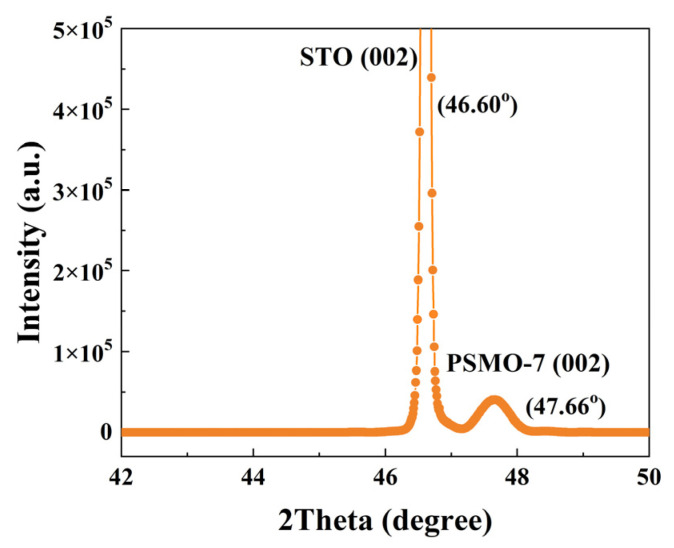
The *ω*-2θ scan of the (002) diffraction plane of the PSMO-7 thin film grown on (001)-oriented SrTiO_3_.

**Figure 2 materials-16-00075-f002:**
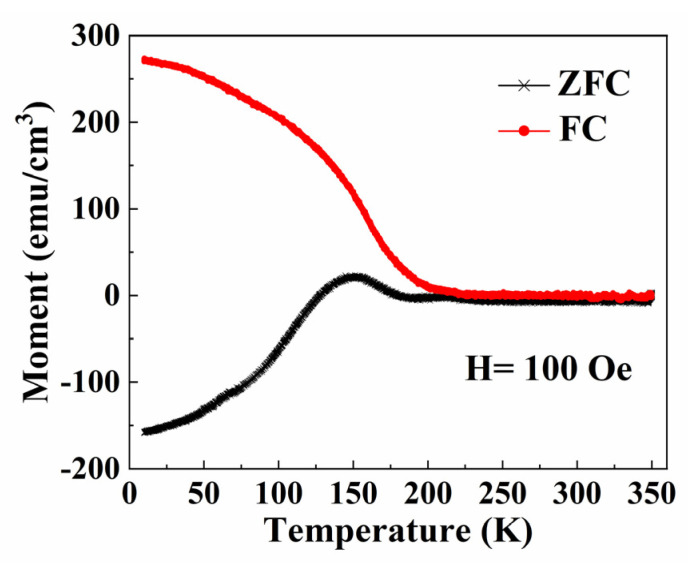
The M(T) data after ZFC and FC processes for PSMO-7 film on SrTiO_3_ under the applied magnetic fields of 100 Oe.

**Figure 3 materials-16-00075-f003:**
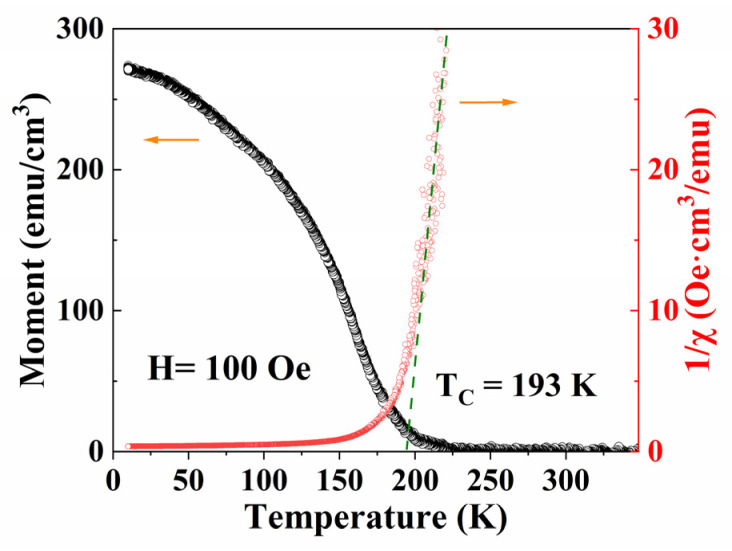
The M(T) and 1/χ(T) curves under FC conditions for PSMO-7 film on SrTiO_3_ at a magnetic field of 100 Oe.

**Figure 4 materials-16-00075-f004:**
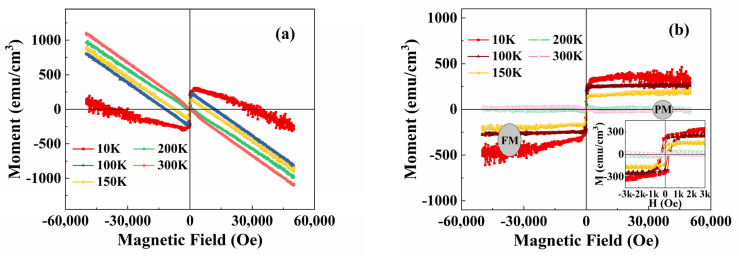
The M(H) loops of the PSMO film on SrTiO_3_ at temperatures of 10, 100, 150, 200, 250, and 300 K. The applied magnetic field changes from −50,000 to 50,000 Oe. (**a**) Represents the magnetic data without subtracting the diamagnetic background of the substrate; (**b**) shows the magnetic data by deducting the diamagnetic background of the substrate. The inset of (**b**) shows the partially enlarged view of the low magnetic field region.

**Figure 5 materials-16-00075-f005:**
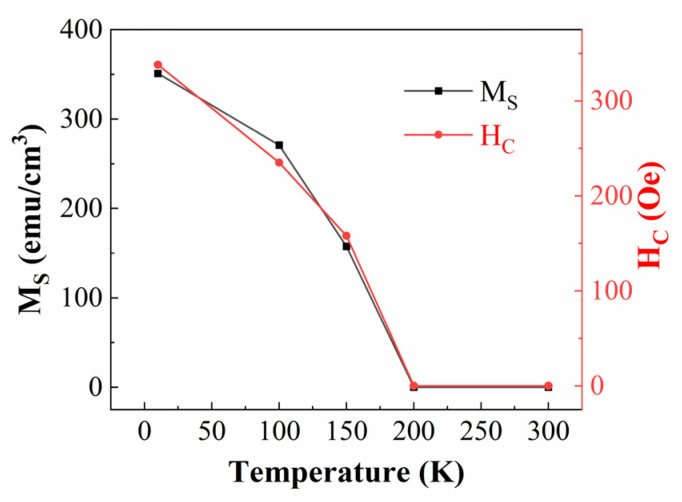
The temperature dependence of saturation magnetization M_S_ and coercive field H_C_ for the PSMO film on SrTiO_3_. Note: the solid lines guide the eyes to see.

**Figure 6 materials-16-00075-f006:**
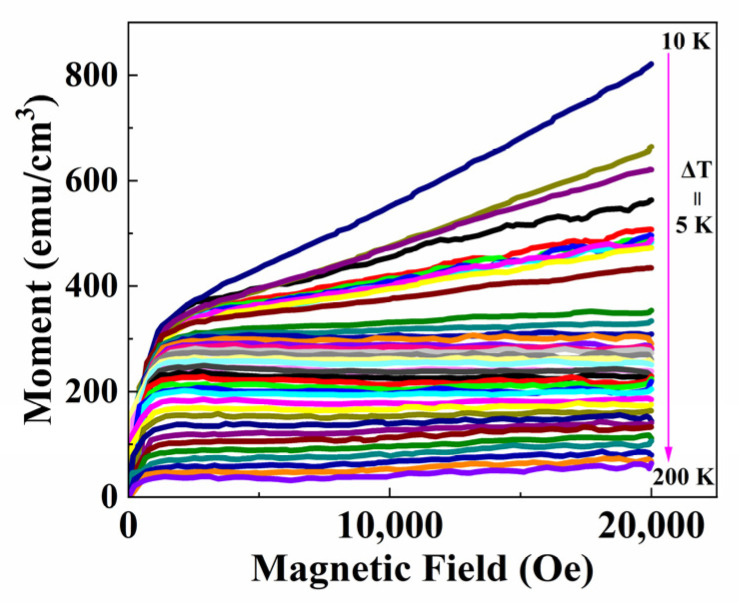
The isothermal magnetization M(H, T) curves with the applied field from 0 to 20,000 Oe measured from 10 to 210 K (with temperature intervals of 5 K) for PSMO film on SrTiO_3_.

**Figure 7 materials-16-00075-f007:**
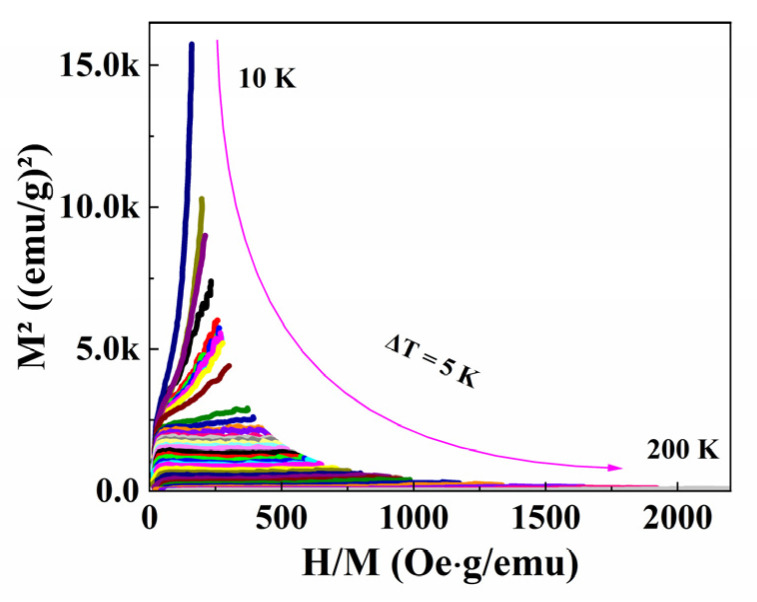
The Arrott plot obtained from the isothermal magnetization.

**Figure 8 materials-16-00075-f008:**
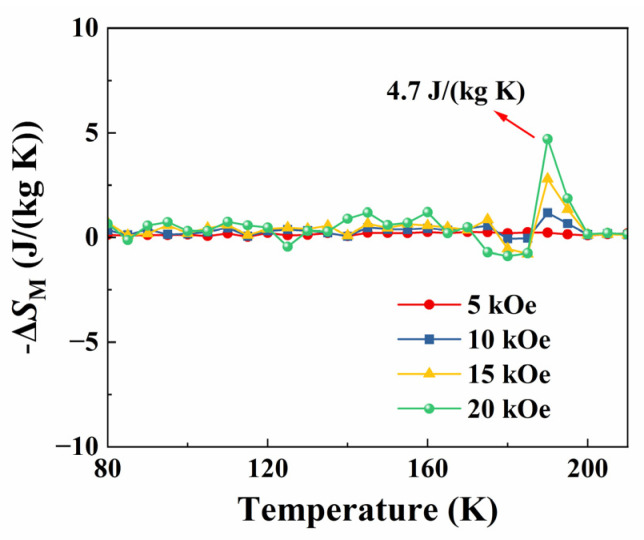
The temperature dependencies of entropy changes ($-\Delta S_{M}$) under different magnetic fields of 5, 10, 15, and 20 kOe for PSMO film on SrTiO_3_.

## Data Availability

All data are available from the corresponding author upon reasonable request.

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
