# Peer review of "The Magnetic Properties and Magnetocaloric Effect of Pr0.7Sr0.3MnO3 Thin Film Grown on SrTiO3 Substrate"

_materials, 2022, doi:10.3390/ma16010075_

Round 1
Reviewer 1 Report
Review Report
The manuscript mainly reports the PLD preparation of Pr0.7Sr0.3MnO3 thin film on SrTiO3 Substrate and studies on the magnetic properties under different temperature. The Pr0.7Sr0.3MnO3 thin film has been found to undergo a phase transition from FM to PM at a temperature of around 193 K, and under 20 kOe, the entropy change of this phase transition has been measured as at 4.7 J/kg×k, which indicates a possible application of Pr0.7Sr0.3MnO3 thin film in the field of magnetic refrigeration. Sufficient results are presented in this manuscript. Also, this manuscript is well written, organized, and argued. Thus, I would recommend the publication of this manuscript on Materials. However, if the introduction part can be polished with more background information, especially the benchmark of the temperature and entropy change of the phase transition, the whole manuscript would be further improved.
Author Response
The manuscript mainly reports the PLD preparation of Pr0.7Sr0.3MnO3 thin film on SrTiO3 Substrate and studies on the magnetic properties under different temperature. The Pr0.7Sr0.3MnO3 thin film has been found to undergo a phase transition from FM to PM at a temperature of around 193 K, and under 20 kOe, the entropy change of this phase transition has been measured as at 4.7 J/kg×k, which indicates a possible application of Pr0.7Sr0.3MnO3 thin film in the field of magnetic refrigeration. Sufficient results are presented in this manuscript. Also, this manuscript is well written, organized, and argued. Thus, I would recommend the publication of this manuscript on Materials. However, if the introduction part can be polished with more background information, especially the benchmark of the temperature and entropy change of the phase transition, the whole manuscript would be further improved.
Answer: Thanks very much for your comments. The background information has been revised in red in the introduction.
Reviewer 2 Report
Comments are as attached.

Reviewer 3 Report
This manuscript describes the magnetic properties of a perovskite type metal oxide ally thin films.
Comments and recommendations:
In the HRXRD pattern, the growth orientation of metal oxide alloys indicated as [001] orientation, how authors determined this.
If these are grown epitaxial orientation to the substrate, why the peak width is wider? Authors should comment on the peak width relating to the crystalites size. How authors comment on the crystal quality, without analyzing the crystal size. Authors should calculate the crystal size from the XRD peak.
Authors discuss the magnetic behavior of the thin films. However, there is no discussion correlating to the prior literature and provided reasoning to why the thin film show such behaviors. Specially temperature dependence saturation magnetization results (Figure 5). Is the drop in magnetization due to crystal size. Crystal domain size plays a huge role in megnetic phase transition and Hc. Authors should provide evidence base explanation to their results. this is a major weakness in this manuscript.
Round 2
Reviewer 3 Report
Authors addressed the recommendations and revised manuscript can be considered for publications.